# Pilot Investigation on the Presence of Anti-Hepatitis E Virus (HEV) Antibodies in Piglet Processing Fluids

**DOI:** 10.3390/ani10071168

**Published:** 2020-07-09

**Authors:** Ilaria Di Bartolo, Luca De Sabato, Eleonora Chelli, Giovanni Loris Alborali, Matteo Tonni, Marina Monini, Alessia De Lucia, Fabio Ostanello

**Affiliations:** 1Department of Food Safety, Nutrition and Veterinary Public Health, Istituto Superiore di Sanità, 00161 Rome, Italy; ilaria.dibartolo@iss.it (I.D.B.); luca.desabato@iss.it (L.D.S.); chelli.eleonora@gmail.com (E.C.); marina.monini@iss.it (M.M.); 2Istituto Zooprofilattico Sperimentale della Lombardia e dell’Emilia-Romagna, 25124 Brescia, Italy; giovanni.alborali@izsler.it (G.L.A.); matteotonni89@gmail.com (M.T.); 3Department of Veterinary Medical Sciences, University of Bologna, Ozzano dell’Emilia, 40126 Bologna, Italy; alessia.delucia3@studio.unibo.it

**Keywords:** Hepatitis E virus, serology, swine, surveillance, processing fluids

## Abstract

**Simple Summary:**

Domestic and wild pigs are the main Hepatitis E virus (HEV) zoonotic reservoirs. Identifying HEV-positive pig farms is important to implement surveillance programs for this emerging zoonotic agent. The aim of this study was to evaluate the use of serosanguineous fluids obtained as part of castration practice (processing fluids (PFs)) to detect anti-HEV antibodies. Ninety-five paired serum and PF samples were collected from newborn piglets of 29 different litters and tested with a commercial ELISA kit. A significant positive correlation (Spearman’s rho: 0.600; *p* < 0.01) was found between the signal-to-cutoff (S/Co) ratio of anti-HEV antibodies in serum and PF samples. In 26 out of 29 litters (89.7%), there was at least one positive piglet in the serum. Sixteen litters out of 29 (55.2%) were also positive in PFs. The detection of anti-HEV maternal-derived antibodies in PFs confirms a past exposure of sows to the virus. PF may represent a rapid, noninvasive and economical tool to identify HEV-positive farms.

**Abstract:**

Identifying Hepatitis E virus (HEV)-positive pig farms is important to implement surveillance programs for this emerging zoonotic agent. The aim of this study was to evaluate the use of serosanguineous fluids obtained as part of castration practice (processing fluids (PFs)) to detect anti-HEV antibodies in newborn piglets. Ninety-five paired serum and PF samples were collected from piglets of 29 different litters and tested with a commercial ELISA kit. A significant positive correlation (Spearman’s *rho*: 0.600; *p* < 0.01) was found between anti-HEV antibodies in serum and PF samples. In 26 out of 29 litters (89.7%), there was at least one positive piglet in the serum. Sixteen litters out of 29 (55.2%) were also positive in PFs. To simulate the use of PF as pooled samples, the limit of detection of the ELISA was assessed mixing the PF sample with strong, medium, medium-weak and weak ELISA titres with 3, 4, 5 and 6 negative PF samples. Our results suggest that it is still possible to identify a positive PF pool when at least one individual PF sample with medium or strong antibody levels is mixed with 5 or 6 individual negative PF samples. The detection of anti-HEV maternal-derived antibodies in PF confirms a past exposure of sows to the virus. PF may represent a rapid, noninvasive and economical tool to identify HEV-positive farms.

## 1. Introduction

The Hepatitis E virus (HEV) is the causative agent of Hepatitis E in humans and animals [1]. HEV is a quasi-enveloped (in blood) or non-enveloped (in feces) single-stranded RNA virus classified in the family *Hepeviridae* and the genus *Orthohepevirus* [2]. *Orthohepevirus A* species includes 8 genotypes (HEV-1 to HEV-8) infecting both humans and other mammalians. The recent definition of HEV subtype reference strains established a set of whole genome reference sequences for the HEV-1 to HEV-8 subtypes of this genus [3,4]. Only HEV-1 to HEV-4 have been detected in Europe. HEV-1 and HEV-2 infect only humans, while HEV-3 and HEV-4 are zoonotic and infect both humans and mammalians.

In Europe, infections by HEV-1 and HEV-2 have been related to travel in endemic areas. However, over the last 10 years, an increasing number of autochthonous infections have been described and linked to the zoonotic transmission of the genotypes HEV-3 and HEV-4 [5] that are recognized as endemic (autochthonous) in some developed regions [6]. Food-borne transmission of HEV-3 and HEV-4 appears to be a major route in Europe, linked to the consumption of raw pork products (mainly liver sausages) and undercooked wild boar meat [7].

The presence of HEV-3, the most common genotype in Europe, has been extensively described in pig populations [8]. The transmission may be favored by the widespread distribution and high prevalence of HEV infection in pig farms.

In Europe, farm-scale HEV seroprevalence ranged from 30% to 98% [9], with differences among countries [10,11,12]. The farm-scale virological prevalence also varies a lot, ranging from 10% to 100% [9]. However, epidemiological data from different studies are difficult to compare due to discrepancies in diagnostic methods and biological matrices (serum, meat juice and feces) used, farm types and pig’s age class examined.

In the last years, detection methods of HEV-RNA and anti-HEV IgG or IgM antibodies have been largely implemented, and now both commercial and homemade tests are available [1,13,14]. A broad real-time reverse transcriptase-polymerase chain reaction (RT-PCR) enables to detect HEV-1 to HEV-4, is used for both animal and human samples and is widely used in most of the studies [15]. The ELISA tests available are mostly based on the capsid protein as an antigen and, since a single serotype has been described so far, can be used for the detection of all genotypes [16].

Because of HEV implications for public health and its widespread diffusion in pigs, it is relevant to determine the prevalence of HEV-positive farms. HEV surveillance programs are required to gain more knowledge about the occurrence and diversity of strains circulating in pig farms and to establish control measures to reduce the risk of HEV infection and transmission [9,17,18].

Castration of swine is used to control aggressive behavior and to improve the taste of pork by eliminating most boar taint. It is expected that suckling piglets receive maternal-derived antibodies (MDA) that could be detected in serosanguineous fluid originating from these tissues, known as processing fluids (PFs).

The objective of this study was to evaluate the use of PF as a biological matrix to detect anti-HEV MDA to assess the serological status of farrow-to-weaning and farrow-to-finish pig farms. Understanding the health status of the pig herds is of paramount importance to implement control measures at the farm level and at the slaughterhouse (i.e., to prevent cross-contamination by separation of pigs supplied by seropositive herds).

## 2. Materials and Methods

### 2.1. Sampling

The study was conducted in two farrow-to-finish herds (A and B, consisting of 2500 and 700 breeders, respectively) with growing pigs located in the same premises in Lombardia Region (Northern Italy). Both farms were Porcine reproductive and respiratory syndrome virus (PRRSV)-positive, stable and characterized by high biosecurity measures. Pig flow followed the one-week batch management production systems, and therefore, castration practice was performed weekly. The European legislation (Council Directive 2008/120/EC) banned the routine tail-docking of pigs; for this reason, in this work, only PF collected from testicles were examined.

Litters before cross-fostering were randomly selected during the castration procedure (approximately 3 days of age). Blood samples and testicles were collected from 3 to 7 randomly selected piglets within a litter, except for 5 litters (from farm A) where only 2 paired samples were analyzed due to sera or PF samples insufficient volume.

Blood samples were collected using a sterile needle for each pig. Testicles were individually collected and transferred into plastic bags. After three hours at room temperature, the tissues were removed and fluids were recovered aseptically. Sera recovered from blood and PF samples were stored at −20 °C until use [19].

Twenty-four litters were selected from farm A. This sample size was calculated to have 95% of probability to detect at least one positive litter when the seroprevalence in sows was ≥12%. In the absence of information on the sensitivity of the ELISA test performed on the PF, a considerably lower expected prevalence value than the prevalence observed by other authors [20] in sows and gilts was used to establish the sample size. Of the 24 examined sows (all born in farms), 6 were primiparous and 18 were pluriparous. This proportion was representative of the age categories of the breeding sows. Due to limited budget, only 5 litters were selected from farm B (2 from primiparous and 3 from pluriparous). No samples were collected from the sows during this study.

### 2.2. Ethical Approval

No ethical approval was required. Processing fluids were collected during routine procedures of boar castrations, and blood samples were collected for veterinary diagnostic purposes (prevalence evaluation of PRRSV viremic piglets at birth). No further direct procedures, besides those mentioned, were specifically carried out on the animals for this study, and therefore, this work did not require ethical approval under Italian law (Decreto Legislativo 4 March 2014, n. 26).

### 2.3. Detection of Anti-HEV Antibodies by ELISA

Total HEV-specific antibodies were detected in sera and PF samples by the species-independent double-antigen sandwich commercial ELISA kit (HEV-Ab, Wantai Biopharmaceutical Inc., Beijing, China), following the manufacturer’s instructions. The commercial ELISA used in this study allows the detection of total anti-HEV antibodies and is validated for serum or plasma but not for PF.

As specified by the manufacturer, the cutoff value was determined as the optical density (OD) mean value of three negative controls plus 0.12. The individual serum or PF samples giving absorbance values greater than the cutoff value were regarded as positive for anti-HEV antibodies. A litter was considered positive when at least one of the serums and/or PF samples collected from piglets resulted in ELISA-positive for MDA anti-HEV.

The signal-to-cutoff (S/Co) ratio was calculated for each examined sample as a ratio between OD and cutoff values (OD/cutoff).

To simulate the use of pooled PF samples, the limit of detection of the ELISA procedure with PF was evaluated by mixing 10 µl of randomly selected PFs with strong, medium, medium-weak and weak ELISA S/Co (respectively: 6.36, 2.53, 1.73 and 1.24) with equal volumes of 3, 4, 5 or 6 negative (S/Co < 0.03) PF samples (dilution from 1:4 to 1:7).

### 2.4. Detection of Anti-HEV IgG Antibodies by Western Blotting

To assess the analytical specificity of ELISA results, 13 randomly selected ELISA-positive PFs were also tested by Western blotting (WB). The samples were selected from those with the highest ELISA OD value and with the highest volume of PF available, allowing to perform the WB test. The HEV capsid protein (lacking the first 111 amino acids, rΔ111ORF2) from a genotype 3 swine HEV strain was expressed by baculovirus in Sf9 insect cells and used as antigen in WB, as previously described. A titration experiment was preliminarily performed to estimate sera dilution to apply in further analysis [14].

The purified ORF2 protein was separated by SDS-PAGE and then transferred to nitrocellulose membrane (Trans-blot transfer medium, Bio-Rad, Segrate, Milano Italy). The membrane was incubated with 5% skim milk in Phosphate-Buffered Saline (PBS) for 3 h. After pre-incubation with a crude protein extract from uninfected Sf9 cells (2 h at 4 °C), serum samples (1:100) were incubated with the HEV capsid protein for 4 h at room temperature. Membranes were then incubated with alkaline phosphatase-conjugated anti-pig IgG (1:12.000; Sigma Aldrich S.r.l., Milano, Italy). The membranes were stained with an anti-swine secondary antibody, anti-whole IgG (H + L chains) (1:15.000; Sigma Aldrich S.r.l., Milano, Italy) conjugated with alkaline phosphatase. Bands were visualized with 1-step NBT (nitro-blue tetrazolium chloride) and BCIP (5-bromo-4-chloro-3’-indolyphosphate p-toluidine salt) solution (Pierce). Serum from a pig tested positive by the ELISA test was used as a positive control (1:100).

### 2.5. Statistical Analysis

Results (individual piglets and litters) were statistically analyzed considering the results of the ELISA test on serum samples as a gold standard.

The proportion of ELISA positive in serum or PF in piglets and litters by farms and by sow parity was compared using Fisher’s exact tests. Confidence intervals were calculated by binomial (Clopper–Pearson) “exact” method based on the β distribution.

To assess a possible correlation between the individual serum and PF S/Co, the nonparametric Spearman’s *rho* correlation coefficient was calculated. However, the correlation coefficient measures the strength of a relation between two variables and not the agreement between them [21]. Therefore, the measurement agreement between serum and PF S/Co was calculated and expressed as limits of agreement [21,22]. The one-sample Kolmogorov–Smirnov (K-S) test was performed to test serum and PF S/Co data distribution. According to the K-S test results, natural Log (LN) transformation was applied to normalize the data. The difference (serum S/Co − PF S/Co), the mean difference (xd) and the standard deviation of the differences (SDd) were calculated for all paired samples. Limits of agreement (LA) were calculated as xd ± 1.96 SDd. The measurement errors on the serum S/Co were assumed nil.

The receiver operating characteristic (ROC) curve was used to assess the optimal cutoff values for S/Co interpretation of the PF results. Sensitivity (Se) and specificity (Sp) were calculated.

Agreement between PF and serum ELISA results using the manufacturer’s cutoff values and the calculated optimal cutoff was assessed for each animal and litter using the Cohen’s Kappa (K) coefficient [23].

Processing fluid pool sensitivity (pSe) was calculated as the ratio between positive litter in serum and positive litter in PFs. According to Christensen and Gardner [24], herd-test sensitivity estimated from pooled samples results (HpSe) was calculated as follow:HpSe = 1 − ((1 − (1 − TP)*^k^* × (1 − pSe) + (1 − TP)*^k^* × (pSp)*^k^*)*^r^*(1)
where TP is the true sero-prevalence, pSe and pSp are pooled test sensitivity and specificity, *k* is the number of individual samples per pool and *r* is the number of examined pools.

Analyses were conducted using SPSS software, version 25 (IBM SPSS, Armonk, NY, USA). 

## 3. Results

### 3.1. Anti-HEV ELISA-Positive Sera and Processing Fluids

Paired sera and PF samples were collected from at least three randomly selected piglets from each of twenty-nine litters.

Ninety-five PF and sera-paired samples from 29 litters were tested for the presence of anti-HEV antibodies. Positive sera and PF samples were detected in both farms. No significant differences (*p* > 0.05) were found in the prevalence of positive litters or positive piglets (in sera or PFs) between the two farms (Table 1).

Overall, 77 sera out of 95 (81.1%; 95% CI: 71.7–88.4) and 38 PFs out of 95 (40.0%; 95% CI: 30.1–50.6) were tested positive for antibodies against HEV. Among the 95 paired samples, 36 were positive for both matrices (Table 2).

Thirteen randomly selected ELISA-positive PFs were also analyzed by WB using the rΔ111ORF2 protein. Eight PFs reacted with the rΔ111ORF2 protein specifically as well as the swine positive control serum confirming the analytical specificity of the ELISA results.

### 3.2. Correlation and Limits of Agreement Between Serum and PF Results

The scatterplot of the S/Co values resulting from the 95 paired serum and PF samples is shown in Figure 1. When the results of anti-HEV antibody level detected in the two sample types were compared using the Spearman’s *rho* coefficient, a positive correlation was observed (Spearman’s *rho*: 0.600; *p* < 0.01). The measurement agreement between serum and PF S/Co was high (Figure 2): 87 of the 95 piglets (91.6%) had S/Co differences inside the limits of agreement (upper LA: 5.58; lower LA: −1.04).

### 3.3. ROC Curve Analysis to Assess the Optimal Cutoff Values for S/Co Interpretation in PF

The ROC curve analysis showed that the best accuracy (area under the curve (AUC): 71.1%) between PF and serum ELISA results occurred when the PF S/Co threshold was ≥0.271 (Figure 3).

### 3.4. Individual and Litter Level ELISA Results

In paired samples, a slight agreement was found (K = 0.19, 95% CI: 0.07–0.32) using the manufacturer’s ELISA cutoff value to compare individual positive sera (77/95) and PF (38/95) samples. Processing fluid results had a Se of 47% (95% CI: 35–58) and Sp of 89% (95% CI: 65–99) when compared with serum samples (Table 2).

Using the calculated cutoff (S/Co threshold ≥ 0.271), PF samples had a Se and Sp of 77% (95% CI: 66–86) and 56% (95% CI: 31–78), respectively, when compared with ELISA results obtained from individual serum samples. The agreement between the PF and serum results was fair (K = 0.27, 95% CI: 0.05–0.47) (Table 2).

Using manufacture cutoff, in 26 out of 29 litters examined (24 from farm A and 5 from farm B), at least one of the sampled piglets was found positive for antibodies against HEV in serum samples. Of these 26 litters positive in sera, 16 had also at least one PF positive sample with a pool sensitivity (pSe) of 62% (95% CI: 41–80) and a pool specificity (pSp) of 67% (95% CI: 9–99). The agreement between the PF and serum results for litters was slight (K = 0.12, 95% CI: 0.00–0.39) (Table 3).

Using the calculated cutoff, in 24 out of 29 litters examined, at least one of the sampled piglets was found positive for antibodies against HEV in PF samples. Of these 24 litters, 23 had also at least one serum positive with a pool sensitivity (pSe) of 88% (95% CI: 70–98) and a pool specificity (pSp) of 67% (95%CI: 9–99). The agreement between the PF and serum results was moderate (K = 0.43, 95% CI: 0.00–0.88) (Table 3).

Assuming pSe of 40%, 60% and 80% and a pool specificity of 65%, the number of PF pools required (*r*) for fixed pool size (*k* = 3) to provide 99% and 95% confidence to detect decreasing values (40%, 30%, 20%, 10% and 5%) of true within-herd sow sero-prevalence (TP) is reported in Table 4. With a pSe of 60%, a minimum of 7, 8 or 9 pools must be tested to provide 99% probability of detecting a prevalence of 30%, 20% or 10% respectively.

No significant differences (*p* > 0.05) were found in the prevalence of positive litters in serum or PF between primiparous and pluriparous sows.

Similarly, no differences were found in the prevalence of positive serum or PF collected from primiparous and pluriparous offspring.

### 3.5. Limit of Detection of the ELISA in PF Pools

Using the manufacturer’s cutoff value, positive pools were obtained when a single positive PF sample (with strong, medium or medium-weak S/Co values) was mixed with up to 4 known negative samples. Pools were all negative when one PF sample with a weak S/Co value was combined with 3 to 6 negative samples (Table 5).

Conversely, all PF pools were positive regardless of the pool size (up to the 1 positive PF diluted with 6 negative PF) when the calculated threshold value was applied (S/Co ≥ 0.271) (Table 5).

## 4. Discussion

The identification of HEV-positive pig farms is the main assumption to implement effective prophylaxis measures for this emerging zoonotic agent and to limit the risk of introducing contaminated products into the food chain. The transmission of HEV between pigs is strongly influenced by environmental fecal contamination, which suggests the possibility to reduce the prevalence of infected pigs by appropriate farm management, hygiene (including effective disinfection of pig housing and equipment between batches) and biosecurity measures [17]. In association with these intra-herd prophylaxis measures, information on the HEV status of the farm would implement preventive measures to slaughter (i.e., pigs from infected farms are slaughtered at the end of the day to prevent cross-contamination).

Processing fluid collection used to screen herds for pig pathogens such as HEV allows for more litters and more pigs to be sampled across farrowing rooms, reducing the handling associated with the blood draw and diagnostic costs compared to multiple individual pig serum samples [25].

Fecal shedding of HEV is limited in time and largely depends on the age of the animals limiting the use of direct diagnostic methods to evaluate the HEV status in pig farms. Detection of antibodies in sera overcomes the described limits but, as a screening method to be applied in farm blood sampling protocols, is cost-prohibitive and not animal friendly.

In pig production, there are management practices (i.e., castration and tail docking) performed during the first week of life that can facilitate the collection of samples that can be used to monitor infection status without representing a major inconvenience [19]. In 2010, a European Declaration on alternatives to surgical castration (e.g., castration with anesthesia or analgesia, immunocastration, raising entire males and sperm sexing) aimed to abandon surgical castration by 2018 was signed. Nonetheless, the target set by the declaration is not met by most European countries yet.

There are several concerns regarding the use of the alternative to surgical castration methods due to the additional costs for farmers and the uncertainties consumer’s attitudes towards meat from pharmacologically castrated pigs. Therefore, it is reasonable to postulate that the complete ban of surgical castration (with or without anesthesia) may not happen in the short-term period. It is likely that, in Europe, PF collection will still be possible for the next 5–10 years. Furthermore, in the other major swine-producing countries (e.g., China, USA and Brazil), the issue of the castration ban has not been raised yet.

The serosanguineous fluids originating from these tissues (PF) can be used as a sensitive sample type to determine the presence of PRRSV [26] or MDA against other pathogens in newborn pigs. Serological examination of the PF has several advantages compared with serum: the collection of the PF reduces discomfort, stress, pain and injury to the animal. Processing fluid samples are also attractive because the sample collection is quick and easy to perform since non-veterinarian staff with training can carry it out.

This study aimed to investigate the use of PF as an alternative screening method for determining the presence of anti-HEV antibodies in farrow-to-finish or farrow-to-weaning farms. Paired PF and serum samples from newborn piglets were analyzed using a commercial ELISA kit to determine the presence of total anti-HEV antibodies. To the best knowledge of the authors, this is the first field study in which PF was used for the detection of anti-HEV MDA. The presence of anti-HEV antibodies in different tissue fluids (meat juice) rather than serum samples had been previously reported [27].

At the individual level, using the manufacturer’s cutoff value to compare the proportion of positive sera and PF samples, a slight agreement (K = 0.19) and a low Se (47%) were found between the two biological matrices. This result is in line with other previous studies in which the antibodies concentration in biological fluids (e.g., oral fluid or meat juice) was found to be much lower compared to sera [22,28,29]. Therefore, it is likely that anti-HEV antibodies concentration in PF is lower than in serum. As already observed for the meat juice [22], PF samples most likely consist of a mixture of serum, lymph and released intracellular liquid and may be regarded as a physiological dilution of serum. Therefore, sera samples with low concentration of antibodies that give a weak OD in the ELISA just above the cutoff may have negative PF results.

Therefore, to adapt the ELISA test designed for sera to be used with PF, the cutoff value for PF samples was calculated using the ROC curve analysis. With this cutoff value, PF samples had Se and Sp values of 77% and 56%, respectively, when compared with ELISA results obtained from individual serum samples. Despite that, the agreement between the PF and serum results were fair (K = 0.27). The level of ingested colostrum antibodies in piglets is dependent on many factors such as the amount of available colostrum, amount of colostrum ingestion, antibodies concentration in sow’s sera, the concentration of colostrum antibodies and their absorption in piglet’s guts, birth order and body mass index [30,31]. All these factors, particularly individual ones (birth order and body mass index), must be taken into account to increase the likelihood of examining PF from piglets with high antibody levels.

No significant difference (*p* > 0.05) was observed in the prevalence of positive litters between gilts and sows, suggesting that the age of the breeding animals might not affect the detection of positive PF.

However, considering that PF should be tested in groups (pools), it is interesting to note that, at the litter level, the results indicate a sufficient pool sensitivity and specificity (88% and 67%, respectively), with a moderate agreement (K = 0.43).

The use of pooled samples allows testing of a large number of animals, reducing time and cost. However, a major limitation to using pools as diagnostic tools is the dilution of positive samples below the limit of detection, resulting in misdiagnosed false negatives [32,33]. Therefore, in the current study, the dilution effect when multiple samples are pooled was investigated.

Our results showed that a positive PF pool is revealed by the ELISA test when at least one individual PF with medium (S/Co > 2.5) or strong (S/Co > 6) anti-HEV antibodies response is mixed with 5 or 6 individual negative PF, respectively. However, this result is largely dependent on the cutoff value. When the ROC curve calculated cutoff value was used, the number of false-negative PF pools decrease and positive PF samples even with higher dilution were detected. All PF samples used to create the artificial pools were positive up to the maximum dilution (1:7). Therefore, assuming that the amount of MDA is also correlated with the sow’s immune level [30], with an average litter size of 14 and a sex ratio of 1:1, a single male pig with a strong S/Co value is sufficient to identify the litter as positive.

Using the calculated PF pool sensitivity (62%, 16 of 29 PF positive litters) and specificity (67%, 2 of 3 negative litters), the required number of pools to detect the presence of HEV seropositive sows within a farm is considerably low. For example, for a PF pool size of 3, a minimum of 7 pools need to be tested to provide 99% probability of detecting a prevalence of 30%.

Our study showed that PF samples can be used for the screening of farrow-to-weaning and farrow-to-finish farms. Antibody detection in PF samples can represent a fast and reliable method to assess the HEV status in pig farms without causing stress and injury to the animals [19].

It is widely acknowledged that the risk of having HEV-contaminated livers at slaughter age is related to two factors: i) coinfection with viruses affecting pig immune response, mainly PRRSV, and ii) a short time-period between HEV infection and slaughter. Intercurrent pathogens may lead to extended HEV shedding, and chronic HEV infection [34] and passive immunity of piglets delayed HEV infection by about six weeks [35]. PF specimens may be a powerful tool to monitor PRRSV and HEV in farrowing houses although this application is limited to male suckling piglets. It is important to note that the collection of this sample type only represents the sow’s immune status and not the piglet’s. Furthermore, considering that colostrum is generally negative for IgM [18], it is possible to detect positive PF only when the HEV infection in the sow took place a few weeks before sample collection. Consequently, PF samples can be used only to highlight HEV infection on farrowing pig farms.

## 5. Conclusions

Results of positivity for antibodies against HEV in PF mean a prior exposure of the animals to HEV, not necessarily corresponding to active shedding of the virus which, even not present in the sow tested, could be still present in the farm. Based on PF surveillance results, the monitoring by molecular assay to detect HEV-RNA could be planned on farms. Indeed, to prevent the risk of introducing infected animals to the slaughterhouse, not only the presence of antibodies against HEV but also the shedding of HEV should be monitored. The detection of anti-HEV antibodies by ELISA in PF samples provides a preliminary result to plan effective prophylactic measures to apply at the farm level.

It is recommended that further large works should be carried out on samples from a larger number of farms.

## Figures and Tables

**Figure 1 animals-10-01168-f001:**
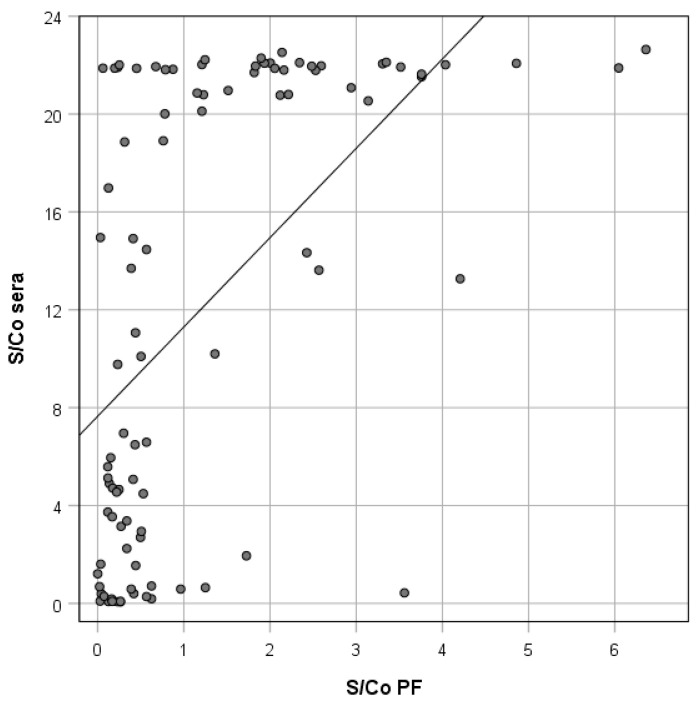
Scatterplot of the signal-to-cutoff (S/Co) ratio values from individual serum and processing fluid (PF) ELISA results in 95 paired samples.

**Figure 2 animals-10-01168-f002:**
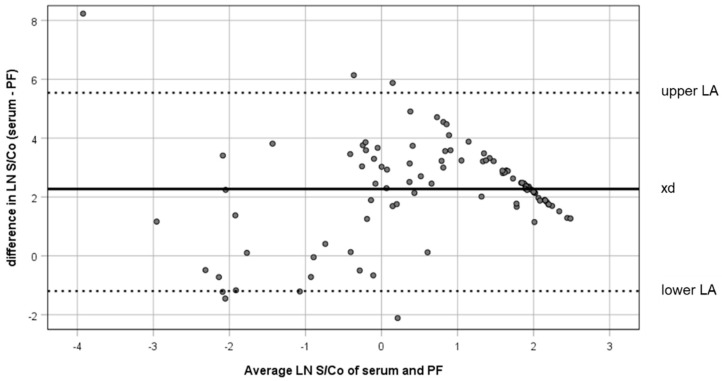
Individual sample comparisons of serum and processing fluid (PF) results: The difference in ELISA S/Co (serum S/Co – PF S/Co) was plotted against the mean S/Co of serum and PF. The mean of difference (xd) and limits of agreement (LA) are marked.

**Figure 3 animals-10-01168-f003:**
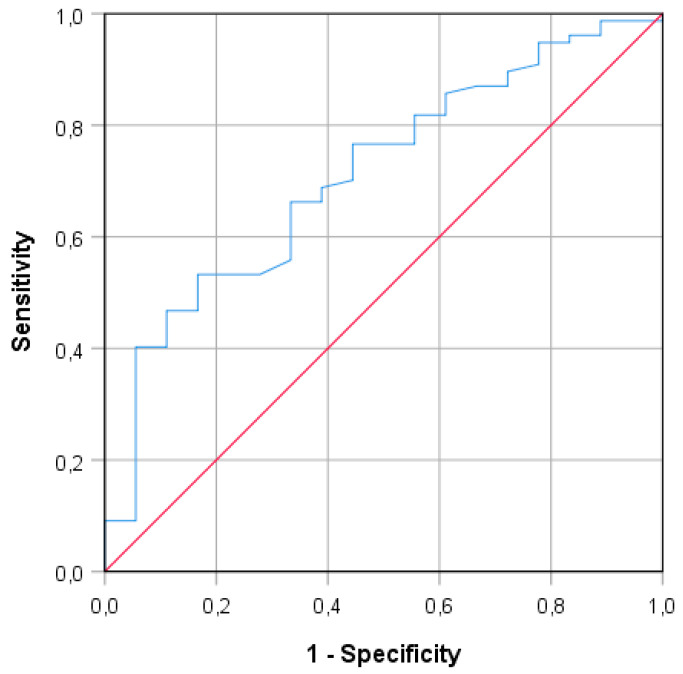
Receiver operating characteristic (ROC) curve for interpreting S/Co ELISA results in processing fluids.

**Table 1 animals-10-01168-t001:** Results of detection of anti-Hepatitis E virus (HEV) ELISA-positive litter in sera and processing fluids (PFs).

Farm	Litter	Positive Litter ^1^ in Sera (No. Positive Piglets/Examined)		Positive Litter ^1^ in PF (No. Positive Piglets/Examined)	
A	1	+	(3/3)	+	(3/3)
A	2	+	(3/3)	+	(2/3)
A	3	+	(2/2)	+	(2/2)
A	4	+	(2/3)	-	(0/3)
A	5	+	(2/3)	-	(0/3)
A	6	+	(3/3)	+	(3/3)
A	7	+	(1/2)	-	(0/2)
A	8	+	(3/3)	+	(1/3)
A	9	+	(2/2)	+	(2/2)
A	10	+	(1/2)	-	(0/2)
A	11	+	(2/3)	+	(2/3)
A	12	+	(3/3)	-	(0/3)
A	13	+	(3/3)	+	(1/3)
A	14	+	(3/3)	+	(2/3)
A	15	+	(2/2)	-	(0/2)
A	16	+	(3/3)	+	(2/3)
A	17	+	(3/3)	+	(3/3)
A	18	+	(3/3)	-	(0/3)
A	19	+	(3/3)	+	(2/3)
A	20	+	(3/3)	+	(1/3)
A	21	-	(0/4)	-	(0/4)
A	22	+	(7/7)	+	(7/7)
A	23	-	(0/5)	+	(1/5)
A	24	+	(3/3)	-	(0/3)
A	total no. positive/ examined	22/24 (91.7%) (CI: 73.0–99.0)	60/74 (81.1%) (CI: 70.3–89.3)	15/24 (62.5%) (CI: 40.6–81.2)	34/74 (46.0%) (CI: 34.3–57.9)
B	25	+	(4/4)	-	(0/4)
B	26	+	(5/5)	-	(0/5)
B	27	-	(0/3)	-	(0/3)
B	28	+	(3/4)	+	(3/4)
B	29	+	(5/5)	+	(1/5)
B	total no. positive/ examined	4/5 (80.0%) (CI: 28.4–99.5)	17/21 (81.0%) (CI: 58.1–94.6)	2/5 (40.0%) (CI: 5.3–85.3]	4/21 (19.1) (CI: 5.5–41.9)
A and B	total no. positive/ examined	26/29 (89.7%) (CI: 72.6–97.8)	77/95 (81.1%) (CI: 71.7–88.4)	17/29 (58.6%) (CI: 38.9–76.5)	38/95 (40.0%) (CI: 30.1–50.6)

^1^ A litter was considered positive when at least one of the serums and/or PFs collected from piglets resulted in ELISA positive for maternal-derived antibodies anti-HEV; 95% confidence intervals (CI) are indicated.

**Table 2 animals-10-01168-t002:** Agreement between processing fluids (PF) and sera-based HEV ELISA results at the individual level.

Cutoff	Anti-HEV Antibodies in PF	Anti-HEV Antibodies in Sera (%)		Total	Cohen’s Kappa
		positive	negative		
Manufacturer’s	positive	36 (94.7)	2 (5.3)	38	
	negative	41 (71.9)	16 (28.1)	57	
	total	77 (90.9)	18 (9.1)	95	
		Se: 47% (CI: 35–58)	Sp: 89% (CI: 65–99)		0.19 (CI: 0.07–0.32)
		positive	negative		
Calculated (≥0.271 S/Co)	positive	59 (88.1)	8 (11.9)	67	
	negative	18 (64.3)	10 (35.7)	28	
	total	77 (81.1)	18 (18.9)	95	
		Se: 77% (CI: 66–86)	Sp: 56% (CI: 31–78)		0.27 (CI: 0.05–0.47)

Se: sensitivity; Sp: specificity; 95% confidence intervals (CI) are indicated.

**Table 3 animals-10-01168-t003:** Agreement between processing fluids (PF) and sera-based HEV ELISA results at the litter level.

Cutoff	Result of Anti-HEV Antibodies per Litter in PF ^1^	Results of Anti-HEV Antibodies per Litter in Sera ^1^ (%)		Total	Cohen’s Kappa
		positive	negative		
Manufacturer’s	positive	16 (94.1)	1 (5.9)	17	
	negative	10 (83.3)	2 (16.7)	12	
	Total	26 (89.7)	3 (10.3)	29	
		pSe: 62% (CI: 41–80)	pSp: 67% (CI: 9–99)		0.12 (CI: 0–0.39)
		positive	negative		
Calculated (≥0.271 S/Co)	positive	23 (95.8)	1 (4.2)	24	
	negative	3 (60.0)	2 (40.0)	5	
	Total	26 (89.7)	3 (10.3)	29	
		pSe: 88% (CI: 70–98)	pSp:67% (CI:9–99)		0.43 (CI: 0.00–0.88)

^1^ A litter was considered positive when at least one piglet had a serum or PF optical density value > ELISA cutoff; pSe: pool sensitivity; pSp: pool specificity; 95% confidence intervals (CI) are indicated.

**Table 4 animals-10-01168-t004:** Number of pools required (*r*) for fixed pool size (*k* = 3) to provide 99% or 95% confidence to detect decreasing values of true within-herd sow prevalence (TP) assuming a pool sensitivity (pSe) of 40%, 60% and 80% and a pool specificity of 65%.

True within-herd Sow Sero-Prevalence (TP%)	pSe: 40%		pSe: 60%		pSe: 80%	
	99%	95%	99%	95%	99%	95%
40	10	7	6	4	4	3
30	10	7	7	5	5	3
20	10	7	8	5	6	4
10	11	7	9	6	8	5
5	11	7	10	7	9	6

**Table 5 animals-10-01168-t005:** The effect of pooling one PF sample with high, medium, medium-low and low ELISA S/Co with three through 6 negative PF samples.

PF ELISA S/Co	Dilution	No. of Positive (Replicates/Total no. of Replicates)	
		Manufacture’ cutoff ^1^	Calculated Cutoff ^2^
strong (S/Co = 6.36)	1:4	1/1	1/1
	1:5	1/1	1/1
	1:6	1/1	1/1
	1:7	1/1	1/1
	total	4/4	4/4
medium (S/Co = 2.53)	1:4	1/1	1/1
	1:5	1/1	1/1
	1:6	1/1	1/1
	1:7	0/1	1/1
	total	3/4	4/4
medium-weak (S/Co = 1.73)	1:4	1/1	1/1
	1:5	1/1	1/1
	1:6	0/1	1/1
	1:7	0/1	1/1
	total	2/4	4/4
weak (S/Co = 1.24)	1:4	0/1	1/1
	1:5	0/1	1/1
	1:6	0/1	1/1
	1:7	0/1	1/1
	total	0/4	4/4
Total	1:4	3/4	4/4
	1:5	3/4	4/4
	1:6	2/4	4/4
	1:7	1/4	4/4
	total	9/16	16/16

^1^ Manufacture’s cutoff: mean neg. optical density (OD) + 0.12; ^2^ Calculated cutoff (≥0.271 S/Co).

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
