# Peer review of "Pilot Investigation on the Presence of Anti-Hepatitis E Virus (HEV) Antibodies in Piglet Processing Fluids"

_animals, 2020, doi:10.3390/ani10071168_

Round 1

Reviewer 1 Report

Comments to the authors

In this work, the authors investigated the processing fluids from routine castration as alternative sample for screening the presence of anti-HEV maternal antibodies in piglets. The work is interesting and it is the first to use such methodological approach. Therefore, in my opinion, the study deserves to be published. However, some aspects need to be improved. Additionally, the text must be grammatically and lexically corrected.

Simple summary

Line 22 and 23: I suggest to add the resulting percentages as well.

Abstract

Line 31: As in the simple summary (line 21), the expression “positive correlation” is misleading. Please specify if it is a statistical measure.

Line 32 and 33: See the comment above.

Line 35 and 37: Generally, number below 10 are written as words. Please check author instructions and eventually correct throughout the text and tables.

Introduction

Line 44: To be more specific, HEV sheds into feces as nonenveloped virions and circulates in the blood in a quasi-enveloped form.

Line 66: Please explain the abbreviation RT-PCR the first time it appears.

Line 75-76: I suggest to add references here. Consider Feng et al. 2011 Infection Dynamics of Hepatitis E Virus in Naturally Infected Pigs in a Chinese Farrow-To-Finish Farm.

Line 75-82: These sentences are not well written. There are excessive subordinate clauses. Please reword.

Materials and Methods

Line 86: Is Northern Italy referred to a single region? In case, please specify.

Line 86-89: As above, the sentence is too long.

Line 101: Are there references to cite here?

Line 104: It is not clear if any samples were taken from adult pigs.

Line 108: See the comment just above.

Line 131: Is there any reason why the authors selected 13 samples?

Line 135: Why was a 1:100 dilution of the sera performed? Is the choice based on the results of the previous ELISA test?

Results

Line 167: A subheading is required here.

Line 169-170: this explanation should be moved in Materials and Methods.

Line 177-179: What about the negative WB sera? Could using non-diluted sera allow to detect more positive samples by WB?

Discussion

Line 286: I agree that the routine castration facilitates the collection of PF. However, PFs are from a surgical method. Thus, in my opinion, it is not appropriate describing the collection as non-invasive.

Line 295-298: This paragraph is an unnecessary repetition of the results.

Line 348: specify what infectious agents the authors refer to. HEV and PRRSV?

The article of Feng et al. 2011 offers some talking points but the authors did not cite that. For instance, colostrum is usually negative for anti-HEV IgM. I wonder if using ELISA kit for specific IgG detection could provide better performances for PF.

Table 1

The caption is confounding. What does “presence of anti-HEV ELISA-positive piglets in paired sera or PF” mean? Please reword.

Please check the text alignment of the first column.

MDA should be written in full here. Check if all non-common abbreviations used in Tables and Figures are explained in the captions or footnotes.

Table 2

Check the consistency of text alignment.

I believe that the way to present the results in Table 2 and 3 can be improved. Is it possible to report the results for PF obtained by manufacturer or calculated cut-off in Table 2 and 3?

Author Response

Comments to the authors

In this work, the authors investigated the processing fluids from routine castration as alternative sample for screening the presence of anti-HEV maternal antibodies in piglets. The work is interesting and it is the first to use such methodological approach. Therefore, in my opinion, the study deserves to be published. However, some aspects need to be improved. Additionally, the text must be grammatically and lexically corrected.

Answer. We appreciate the time and effort that the reviewer has dedicated to our manuscript. The comments have identified several aspects that required improvement; therefore, we are grateful to the reviewer for insightful suggestions. As required, the text has been grammatically and lexically modified in various parts of the manuscript.

Simple summary

Line 22 and 23: I suggest to add the resulting percentages as well.

Answer. We agree with the Reviewer’s comment as suggested the resulting percentages are now included in the Simple summary (line 23) and in the Abstract (lines 34, 35).

Abstract

Line 31: As in the simple summary (line 21), the expression “positive correlation” is misleading. Please specify if it is a statistical measure.

Answer. We agree with the Reviewer’s comment and the following sentence has been rephrased accordingly to the statistical analysis results: “A significant positive correlation (Spearman's rho: 0.600; p<0.01) was found between signal-to-cut-off (S/Co) ratio of anti-HEV antibodies in serum and PF samples” (line 21).

Line 32 and 33: See the comment above.

Answer. Please See the answer above. The sentences (lines 32, 33) were modified to address the Reviewer’s comment.

Line 35 and 37: Generally, number below 10 are written as words. Please check author instructions and eventually correct throughout the text and tables.

Answer. We have carefully checked the author instructions and we have not found any indications on this point. Consequently, the text and tables have not been changed.

Introduction

Line 44: To be more specific, HEV sheds into feces as nonenveloped virions and circulates in the blood in a quasi-enveloped form.

Answer. To address the Reviewer’s comment the sentences (line 46) was rephrased.

Line 66: Please explain the abbreviation RT-PCR the first time it appears.

Answer. The explanation of the abbreviation was added (line 68).

Line 75-76: I suggest to add references here. Consider Feng et al. 2011 Infection Dynamics of Hepatitis E Virus in Naturally Infected Pigs in a Chinese Farrow-To-Finish Farm.

Answer. We thank the Reviewer for the suggestion. The new reference (no. 18) was added to the manuscript (line 75).

Line 75-82: These sentences are not well written. There are excessive subordinate clauses. Please reword.

Answer. To address Reviewer’s comments these two sentences were revised (lines 80-88).

Materials and Methods

Line 86: Is Northern Italy referred to a single region? In case, please specify.

Answer. We agree with the Reviewer. The region (Lombardia, Northern Italy) has been specified (line 91).

Line 86-89: As above, the sentence is too long.

Answer. The authors agree with the Reviewer’s comment. The sentence was modified (lines 93-97).

Line 101: Are there references to cite here?

Answer. As suggested, the following reference (Vilalta et al, 2018. Use of processing fluids and serum samples to characterize porcine reproductive and respiratory syndrome virus dynamics in 3 day-old pigs. Vet Microbiol, 225, 149-156; ref. no. 19.) was added to the manuscript (line 104).

Line 104: It is not clear if any samples were taken from adult pigs.

Line 108: See the comment just above.

Answer. No samples were taken from the sows. To clarify this, a sentence was added (line 112).

Line 131: Is there any reason why the authors selected 13 samples?

Answer: The main reason why we decided to carry out Western blotting on PF samples was to assess the analytical specificity of the ELISA kit. For this only purpose and with a limited number of PF samples with sufficient volume we have chosen a few samples among the positive ELISA PF samples to test and the results we got confirmed the analytical specificity of the ELISA results.

The same antigen was successfully used to test antibodies in sera. Since we have mostly observed a lower OD values in the ELISA test with PF samples than with sera, we used a low dilution 1:100 to perform the WB with PF. Therefore, the 13 samples selected were those with the highest OD value in ELISA test (to guarantee that at 1:100 we would have observed a staining of the antigen) and with the highest quantity of PF allowing to perform the WB tests (lines 138-139).

Line 135: Why was a 1:100 dilution of the sera performed? Is the choice based on the results of the previous ELISA test?

Answer. The choice is based on our previous study (Ponterio et al., 2014; ref. no. 14) where we had observed the best dilution of sera to have highest specificity and sensitivity was from 1:100 to 1:200. As revealed by titration experiment (Ponterio et al., 2014) which enables the selection of the sera dilution which gave best staining and minimum background. In the present study, we used the same antigen and since we have frequently observed a lower OD values in the ELISA test with PF samples than with sera, we used the lower dilution 1:100 to perform the WB with PF.

Results

Line 167: A subheading is required here.

Answer. To address the Reviewer’s comment a subheading (“3.1. Anti-HEV ELISA-positive sera and processing fluids”) was added (line 181). Consequently, the other subheading has been renumbered.

Line 169-170: this explanation should be moved in Materials and Methods.

Answer. The authors agree with the Reviewer's suggestion and the paragraph has been moved to the Materials and Methods section (Line 100).

Line 177-179: What about the negative WB sera? Could using non-diluted sera allow to detect more positive samples by WB?

Answer. Probably yes, but for most of the samples it was not feasible because a limited amount of PF was available. We had observed similar results, some negative results in WB not corresponding to the ELISA, in previous studies conducted using the same antigen to assess the presence of antibodies anti-HEV in sera from rabbits and pigs. These previous finding corroborates the results obtained in this study by WB which has only been used to confirm the specificity of ELISA since we had not previous experiences with the PF.

Discussion

Line 286: I agree that the routine castration facilitates the collection of PF. However, PFs are from a surgical method. Thus, in my opinion, it is not appropriate describing the collection as non-invasive.

Answer. We agree with the Reviewer's comment and therefore the term "non-invasive" has been deleted (line 306).

Line 295-298: This paragraph is an unnecessary repetition of the results.

Answer. We agree with the Reviewer's comment and that paragraph was deleted (lines 315-318).

Line 348: specify what infectious agents the authors refer to. HEV and PRRSV?

The article of Feng et al. 2011 offers some talking points but the authors did not cite that. For instance, colostrum is usually negative for anti-HEV IgM. I wonder if using ELISA kit for specific IgG detection could provide better performances for PF.

Answer. To address the Reviewer's comment, the following changes were made: the text was modified specifying that the authors refer to PRRSV and HEV (line 369). Moreover, some points from the work of Feng et al. 2011 were also discussed (lines 372-374).

We thank the Reviewer for the suggestion regarding the type of ELISA kit. We used an ELISA kit for total antibodies to increase the probability of detecting positive PF. Also Feng et al. 2011 reported that at 0 days of life 12.5% of the subjects were positive for total antibodies while none were positive for anti-HEV IgG or anti-HEV IgM.

Table 1

The caption is confounding. What does “presence of anti-HEV ELISA-positive piglets in paired sera or PF” mean? Please reword.

Answer. We agree with the Reviewer. The caption was rewritten. “Results of detection of Anti-HEV ELISA-positive litter in sera and processing fluids (PF)”.

Please check the text alignment of the first column.

MDA should be written in full here. Check if all non-common abbreviations used in Tables and Figures are explained in the captions or footnotes.

Answer. We agree with the Reviewer. Text alignment, abbreviation, captions, and footnotes in tables and figures have been modified accordingly.

Table 2

Check the consistency of text alignment.

I believe that the way to present the results in Table 2 and 3 can be improved. Is it possible to report the results for PF obtained by manufacturer or calculated cut-off in Table 2 and 3?

Answer. We agree with the Reviewer. Tables 2 and 3 have been improved.

Reviewer 2 Report

The manuscript that I reviewed “Pilot investigation on the presence of anti – Hepatitis E Virus (HEV) antibodies in piglet processing fluids” is a study aimed to evaluate the opportunity to use processing fluids obtained during castration practice in order to detect anti- HEV maternal derived antibodies in newborn piglets, representing a rapid, non-invasive and economical tool to identify HEV positive farms.

Major comments

The manuscript is well written and supported by the presented data. PF samples could be a good matrix, rapid and non-invasive, for the purpose indicated but at the same time, in my opinion, the use of this kit is not very economical. Furthermore, I think that it should be better explained the meaning of serological positivity of the farms tested since to detect antibodies is not necessarily associated to the concomitant circulation of the virus but it could be due to previous infections of sows. For this reason and also for the high number of swine seropositive farms, as reported in literature, it will be more useful to perform also a molecular assay in order to identify the farms whose pigs must be slaughtered at the end of the day to prevent cross-contamination.

Minor comments

1)Line 115, the test used is a species‐independent commercial double antigen sandwich ELISA kit? I suggest to the Authors to explain more in deep the type of ELISA employed.

 2) Line 130, “2.4. Detection of anti-HEV IgG antibodies by Western blotting”: I suggest to the Authors to better explain this procedure. They selected ELISA positive PF samples to test in Western Blotting but They didn’t explain how They performed the test since They indicated only the dilution of sera. Furthermore, the Authors detected total antibodies in ELISA and how did They select the IgG positive sera or PF samples? Can You add a figure of the WB?

3)Line 187, Table 1: In my opinion the table is not very clear or there are some inconsistences since several litters were considered positive for sera or PF despite the piglets tested were negative. For example, in litter 5, 7, 10, 12 and others.

4)Line 204, Table 2, I suggest to the Authors to better develop the table, since is not clear. Anti-HEV antibodies in PF should be referred to the right column. Furthermore, I think that the number 75 for the total negatives is wrong. The correct number should be 57.

5)Line 218, Is 17 the correct number of the positive PF litters?

Author Response

Comments and Suggestions for Authors

The manuscript that I reviewed “Pilot investigation on the presence of anti – Hepatitis E Virus (HEV) antibodies in piglet processing fluids” is a study aimed to evaluate the opportunity to use processing fluids obtained during castration practice in order to detect anti- HEV maternal derived antibodies in newborn piglets, representing a rapid, non-invasive and economical tool to identify HEV positive farms.

Answer. We appreciate the time and effort that the Reviewer has dedicated to our manuscript.

Major comments

The manuscript is well written and supported by the presented data. PF samples could be a good matrix, rapid and non-invasive, for the purpose indicated but at the same time, in my opinion, the use of this kit is not very economical. Furthermore, I think that it should be better explained the meaning of serological positivity of the farms tested since to detect antibodies is not necessarily associated to the concomitant circulation of the virus but it could be due to previous infections of sows. For this reason and also for the high number of swine seropositive farms, as reported in literature, it will be more useful to perform also a molecular assay in order to identify the farms whose pigs must be slaughtered at the end of the day to prevent cross-contamination.

Answer. we agree with the reviewer that serology indicates prior exposure to HEV and the animals may not be still shedding the virus at the time of sampling.

We agree that serology most frequently indicates prior exposure to HEV, which could occur in sow before being introduced in the farm or at younger age but to our knowledge the virus if introduced in the farm persists for long time. Furthermore, assessing the serological status of the farm by testing the sows could be considered a first approach to select which farm should be further tested by molecular assays for HEV-RNA detection. Indeed, lower number of sera or PF needs to be tested compared to molecular assays, since a higher percentage of seropositive is expected compared to HEV virological positivity.

This concept is further discussed in the Discussion section (lines 282-285) and also in the Conclusion section (lines 378-383). A new reference (no. 25) was added.

Minor comments

1) Line 115, the test used is a species‐independent commercial double antigen sandwich ELISA kit? I suggest to the Authors to explain more in deep the type of ELISA employed.

Answer. We agree with the Reviewer's suggestion. The description of the ELISA kit employed has been included in the text (lines 120-121).

2) Line 130, “2.4. Detection of anti-HEV IgG antibodies by Western blotting”: I suggest to the Authors to better explain this procedure. They selected ELISA positive PF samples to test in Western Blotting but They didn’t explain how They performed the test since They indicated only the dilution of sera. Furthermore, the Authors detected total antibodies in ELISA and how did They select the IgG positive sera or PF samples? Can You add a figure of the WB?

Answer. We have selected the samples to be tested in WB by volume available (since a limited quantity was available) and higher OD values obtained by ELISA. We cannot exclude that PF tested negative in WB and positive in ELISA would have other classes of immunoglobulins against HEV not stainable by WB (lines 138-139). The test was performed as described in Ponterio et al., 2014 (ref. no. 14).

The ORF2 protein was separated by SDS-PAGE, and then transferred to nitrocellulose membrane (Trans-blot transfer medium, Bio-Rad). Subsequently aspecific binding were prevented by blocking the membrane with 5% skim milk in PBS. Then the membrane was incubated with sera (diluted 100) including positive control (ELISA positive serum) in PBS containing 0.05% Tween-20 and 1% skim milk, for 4 hours. Membranes were then incubated with alkaline phosphatase-conjugated anti-pig IgG (1:12.000; Sigma Aldrich S.r.l., Milano, Italy). Bands were visualized with 1-step NBT/BCIP solution (Pierce) (lines 147-152).

Only after the western was performed it was possible to establish the IgG isotype of the antibodies as the secondary antibody used was an anti pig IgG.

We can provide the figure of WB for the reviewer we do not consider to include the picture in the paper, since in our opinion will not add any values to our results. The main goal of this study was the use of PF in ELISA, the WB was used to confirm that also by PF (we already know for sera) the same antigen would have been stained

3) Line 187, Table 1: In my opinion the table is not very clear or there are some inconsistences since several litters were considered positive for sera or PF despite the piglets tested were negative. For example, in litter 5, 7, 10, 12 and others.

Answer. We apologize for the mistake. Table 1 has been improved and the values have been corrected.

4) Line 204, Table 2, I suggest to the Authors to better develop the table, since is not clear. Anti-HEV antibodies in PF should be referred to the right column. Furthermore, I think that the number 75 for the total negatives is wrong. The correct number should be 57.

Answer. We agree with the Reviewer. Tables 2 and 3 have been improved. As noted by the Reviewer, the total number of negative is 57. We apologize again for the mistake.

5) Line 218, Is 17 the correct number of the positive PF litters?

Answer. Overall, using manufacture cut-off, the number of the PF-positive litters is 17 and the number of serum-positive litters is 26. Sixteen out of 26 serum-positive litters were positive also in PF. To avoid confusing, the phrase was changed (line 237).